



# JULES-GL7: The Global Land Configuration of the Joint UK Land Environment Simulation version 7.0

r860

Andrew J. Wiltshire[1], Carolina Duran Rojas[2], John Edwards[1], Nicola Gedney[1], Anna B. Harper[2], Andy
Hartley[1], Maggie Hendry[1], Eddy Robertson[1], Kerry Smout-Day[1]

[1]Met Office, Fitzroy Road, Exeter. EX1 3PB. UK
[2]University of Exeter, Exeter, UK.

*Correspondence to*: Andrew J. Wiltshire (andy.wiltshire@metoffice.gov.uk)

**Abstract.**

We present the latest global land configuration of the Joint UK Land Environment Simulator (JULES) model as used in the latest international coupled model intercomparison project (CMIP6). The configuration is defined by the combination of switches, parameter values and ancillary data, which we provide alongside a set of historical forcing data that defines the experimental setup. In addition, we provide a standardised modelling system that runs on the NERC JASMIN cluster accessible to all with links to JULES. This is provided so that users can test and evaluate their own science against the standard configuration to promote community engagement in the development of land surface modelling capability through JULES. It is intended that JULES configurations should be independent of the underlying code base and thus they will be available at the latest release of the JULES code. This means that different code releases will produce scientifically comparable results for a given configuration version. Versioning is therefore determined by the configuration as opposed to the underlying code base.





## 1 Introduction

The Joint UK Land Environment Simulator (JULES) (Best et al., 2011; Clark et al., 2011) is the land surface model used by the UK land, hydrological, weather and climate communities. JULES is a comprehensive model simulating the atmospheric exchange of radiation, heat, water, momentum, carbon and methane and changes in the surface states of moisture, heat and

carbon. All these processes are important for the wide ranging application of JULES from carbon cycle (Le Quéré et al., 2018) to climate impact (Shannon et al., 2018) and hydrological (Betts et al., 2018) modelling. However, each of these applications is best suited to a combination of different processes and schemes, for instance an interactive dynamic vegetation model is important to understanding carbon cycle processes but not crucial to crop modelling (Osborne et al., 2015) and may introduce additional biases and errors. The JULES code base enables a vast number of different setups through parameter and switch

combinations, many of which are undesirable for a plethora of reasons from poor performance, lack of testing or incompatibility between options. This can lead to very poor scientific outcomes if the user is not completely familiar with the JULES code base. Addressing this is best achieved by having defined science configurations, specifying a particular combination of parameters and switches that are known to produce appropriate, well evaluated and tested results. JULES is the land component of the Met Office modelling system, which is used across weather to climate timescales. Each component

has defined configurations: the Global Atmosphere (GA) for configurations of the atmospheric model, Global Land (GL) configuration for JULES, and likewise for the ocean and sea-ice components, with the Global Coupled (GC) configuration for the fully coupled atmosphere, ocean, sea-ice and land model. Here, we present the JULES-GL7.0 and JULES-GL7.2 configurations developed primarily as part of the atmosphere model at the Met Office. JULES-GL7.0 is the offline version of the GL7.0 configuration used in conjunction with GA7.0 (Walters et al., 2017), the atmospheric configuration of the Met

Office. These are the latest iterations in the GA/GL configuration series developed for use in global modelling and underpin the HadGEM3-GC3.1 (Williams et al., 2018) model that is being used as part of the sixth iteration of the Coupled Modelling Intercomparison Project (CMIP6) (Eyring et al., 2016). The GL7.0 configuration is specifically developed to simulate the exchange of heat, water and momentum generally known as the 'physical environment,' and therefore does not include biogeochemical components which come under 'earth system' modelling, neither does it include processes specifically related

to climate impacts such as crops. It is the appropriate configuration for understanding hydrology and land surface processes relating to the partitioning of heat and radiation. In many ways, GL7.0 is the core JULES configuration as, for instance, the earth system setup adds components to it to enable the simulation of the exchange of carbon and methane.

Here, we document the offline JULES-GL7 configurations and their release at JULES vn5.3. The release includes a

standardised suite control setup to initialise, reconfigure, spinup and run a standard experiment. The release is designed to be as easy as possible to access and run on the NERC JASMIN platform (http://www.jasmin.ac.uk/). The configuration management makes use of Rose and Cylc suite management tools (https://metomi.github.io/rose/doc/html/index.html). These can be installed locally but are available and maintained on JASMIN, which is the recommended platform for most users.

As a configuration of the JULES model, JULES-GL7 will be available at subsequent model versions and tested to ensure the setup produces scientifically comparable results between model versions until a date when JULES-GL7 is superseded and retired. It is recommended that users use the latest version of the code base to benefit from bug fixes, ease of testing and implementing developments, which can only take place at the head of the JULES code trunk. All bug fixes, which would otherwise affect the scientific comparability of configurations, are implemented on a temporary logical switch so scientific

comparability is maintained over different code versions. These temporary switches are reviewed periodically and are retired when they are no longer required. Living documentation of the latest suite version can be found on the JULES trac configuration page (https://code.metoffice.gov.uk/trac/jules/wiki/JulesConfigurations) and under ticket #837





(https://code.metoffice.gov.uk/trac/jules/ticket/837). Model developers should use the suite and information presented here in combination with the JULES technical documentation found in Best et al., (2011) and Clark et al., (2011).

The JULES suites are available via the Met Office Science Repository Service (MOSRS)
(https://code.metoffice.gov.uk/trac/home) and are freely available subject to completion of a software licence (see Section 6 for details).

JULES-GL7              https://code.metoffice.gov.uk/trac/roses-u/browser/b/b/3/1/6

JULES-GL7.2          https://code.metoffice.gov.uk/trac/roses-u/browser/b/b/5/4/3

### 1.2 Land Configurations in use in Hadley Centre Models

Although this is the first time a standalone standard configuration of JULES is being made available to the community, land configurations are widely established in weather and climate modelling. The predecessor to JULES was the Met Office Surface
Exchange Scheme (MOSES, Cox et al., 1999). Configurations of MOSES2.2 (Essery et al., 2003) underpin the CMIP5 physical model, HadGEM2 (Martin et al., 2011) and the earth system model HadGEM2-ES (Collins et al., 2011). As of GL3 (Walters et al., 2011), configurations of JULES were introduced and have been developed over subsequent iterations of the model development cycle. The latest of which is GL7 as described here offline and as coupled to the atmosphere (Walters et al., 2017) and ocean (Williams et al., 2018). Future configurations are currently in development with the aim of reducing model biases
and improving the representation of physical processes. For instance, GL8 is scheduled to include some updated snow process representation including a new scheme parameterising snow grain size growth (Taillandier et al., 2007) with the aim of reducing albedo biases over the Antarctic and Greenland ice sheets.

### 2 JULES-GL7 Configuration

This section describes the offline JULES-GL7.0 and JULES-GL7.2 science configurations. An important difference between
the offline and coupled versions is that in the coupled version JULES acts as an interface between land (including land-ice), sea-ice and the ocean whilst offline only land is considered. Important parameters are listed in Tables 1 and 2, ancillaries in Table 3 and in most cases switches are listed in the text. The full set of switch settings can be found in the rose suites. We include the full parameter tables for the surface tiles here for clarity as the rose suites namelists encompass all parameters in JULES many of which are linked to particular switches and options and therefore not used in JULES-GL7.0. The appropriate
JULES documentation papers remain Best et al. (2011) and Clark et al. (2011). Where new developments are included, they are described in more detail with appropriate references herein.

### 2.1 Surface Tiling

JULES-GL7 uses a surface tiling scheme to represent subgrid heterogeneity. Within a gridbox, each tile has its own surface energy budget and is coupled to a single shared soil column (Figure 1). Each tile therefore has its own albedo, surface
conductance to moisture, turbulent fluxes, ground heat flux, radiative fluxes, canopy water content, snow mass and melt, and thus surface temperature. Each tile requires its own parameter set which are given in Tables 1 (non-vegetated surface types) and 2 (vegetated surface types) and spatially explicit parameters in Table 3.





There are nine surface tiles consisting of five Plant Functional Types (PFTs) (Broadleaf trees, Needleleaf trees, C3 grass, C4 grass, Shrubs) and four non-vegetated surface types (Urban, Inland Water, Bare Soil and Ice) (Figure 2). These can co-exist in the same gridbox except for ice. The C4 distinction reflects a different photosynthetic pathway with all other PFTs represented as C3. The tile fractions are spatially varying and are read from an ancillary file. The fractions are produced by a re-mapping

5 of the 17 surface types in the International Geosphere-Biosphere Programme (Loveland and Belward, 1997) to the 9 surface types in JULES. The landcover class remapping procedure is described in Table 4 of Walters et al. (2018) and the cross-walking table relating land-cover classes to PFTs in Table B1.

### 2.1.1 Plant Functional Types (PFTs)

Vegetation is represented by the five plant functional types described above. In JULES-GL7 each PFT has its own energy budget including thermal heat capacity (*CanMod=4*), which is a function of the PFT height (2.1.1.1 and 2.3). Leaf level stomatal conductance and photosynthesis are coupled through $CO_2$ diffusion with PFT specific parameters controlling sensitivity to humidity deficit and internal to external $CO_2$ pressures (Cox et al., 1998 and Table 2; *f0, dqcrit*). This coupling

implies that both the energy and carbon cycles are closely related; rising atmospheric $CO_2$ influences stomatal conductance and therefore the surface energy budget. This mechanism is known as physiological forcing (Betts et al., 2007; Field et al., 1995; Sellers et al., 1996). Leaf level conductance must be scaled to the canopy level and in JULES-GL7 this is done using a 10 layered canopy approach (*CanRadMod =4*). At each level separate direct and diffuse Photosynthetically Available Radiation (PAR) levels are calculated using the two-stream approach (Sellars, 1985) to give a profile of PAR through the

canopy. From this the leaf level photosynthesis can be calculated using PFT specific parameters combined with the Collatz (1992, 1991) leaf biochemistry model utilising separate mechanisms for C3 and C4 plants (Jogireddy et al., 2003; Mercado et al., 2007). At each level, if net photosynthesis is negative or stomatal conductance is below a minimal threshold (*glmin*, Table 1), the stomata are closed and the stomatal conductance is set to this minimum. A further mechanism scales leaf level conductance via photosynthesis according to the availability of soil moisture in the rooting profile. In JULES-GL7, this scalar

(β) relates the rooting profile (*rootd,* Table 2) in each soil layer with the availability of soil moisture. β is a piecewise function that scales from 0 when soil moisture is at or below the *wilting point* to 1 where soil moisture is above the *critical point* (Eq 12; Best et al., 2011). The root fraction weighted (*fsmc_mod=0*) total across soil layers value of β is used to scale photosynthesis at the leaf level. Canopy conductance is the leaf area weighted sum of leaf conductance across the 10 levels. A direct output from this setup is a diagnostic of Gross Photosynthetic Potential (GPP).

### 2.1.1.1 Spatial Leaf Area and Canopy Height ancillary data

Leaf Area Index (LAI) is defined as the one-sided surface area of canopy leaf cover per unit area of land and is defined spatially and temporally for each vegetated surface tile in JULES-GL7. Similarly, canopy height is spatially varying per vegetated tile but fixed in time. The ancillaries are derived from satellite data processed to be consistent with the landcover and plant functional type classifications used in JULES-GL7. To do this requires decomposing a 'mixed' signal from the satellite data

into individual PFT contributions. This is achieved via an additional parameter, the "balanced" LAI (*Lb*), meaning the LAI that would be reached if the plant was in full leaf (Table B2). The combination of the mapping from land cover classes to PFTs and the balanced LAI weighting per PFT per land cover class allows the observed gridded satellite value to be decomposed into individual PFT contributions.





In JULES-GL7, monthly variations in LAI are based on a climatology for the period 2005 to 2009 derived from the MODIS LAI product (MOD15; (Yang et al., 2006)). The LAI value for a given PFT, land cover class and month are calculated as follows:

$$LAI_{i,j} = LAI_{MODIS} \frac{(Lb_{i,j}\alpha_{i,j})}{\sum_i(Lb_{i,j}\alpha_{i,j})}$$

Equation 1

Where $LAI_i$ is the LAI for PFT class $i$, $LAI_{MODIS}$ is the MODIS LAI value for a given month, $Lb_{i,j}$ is given by the LAI lookup table (Table B2), $\alpha_{i,j}$ is the fraction of each PFT $i$ in IGBP class $j$ given by the lookup table in Table B1. The PFT-specific LAI ($LAI_i$) is accumulated for all land cover classes in a gridbox for a given month. The resulting input ancillary is then internally

interpolated within JULES to each model timestep. The seasonally-varying LAI for 5 PFTs for 30-60°N is shown in Figure 3. An outcome of this approach is that JULES is forced with the snow-free LAI which explains the large winter reductions in LAI for needleleaf trees.

The introduced balanced LAI has the property of being allometrically related to the canopy height. Based on this allometric

relationship the canopy height ($H$) can be derived for each PFT in each landcover class:

$$H_{i,j} = h_i Lb_{i,j}^{\frac{2}{3}}$$

Equation 2

Where $h_i$ is a PFT specific scalar given in Appendix B (Table 7). The PFT cover mean height (*canht*) is the area weighted

arithmetic mean of the landcover classes in that gridbox.

### 2.1.2 Non-vegetated surface types

The four non-vegetated surface types (urban, inland water, bare soil and land ice) are represented as tiles with separate energy balances, described using the parameters listed in Table 1. A full description of the representation of the non-vegetated surface

types can be found in Best et al. (2011) and the developments since this paper have been highlighted here. Since GL3.0 (Walters, et al., 2011) the urban surface has been represented by the simple one-tile scheme (*l_urban2t=.false.*), which consists of a radiatively coupled (*vf*, Table 1) "urban canopy" with the thermal characteristics (*ch*, Table 1) of concrete (Best, 2005). The urban canopy has a capacity to hold water (*catch*, Table 1) and when wet the surface moisture resistance is reduced to zero. Similar to the urban surface, lakes are represented as a radiatively coupled "inland water canopy" with the thermal

characteristics of a mixed layer depth of water ($\approx$ 5m). The original representation of inland water, as a freely evaporating soil surface (*ch* = *vf* = 0.0, Table 1), was shown to have incorrect seasonal and diurnal cycles for surface temperatures and therefore evaporation (Rooney and Jones, 2010). The high thermal inertia of the urban and lake tiles results in an improved diurnal cycle in surface air temperature. Bare soil or bare-ground surface types are represented as having no canopy heat capacity and a surface moisture resistance to evaporation as a function of surface soil moisture (Eq 17, Best et al., 2011). Ice surfaces are an

exception to the representation of surface heterogeneity as only ice can exist in a gridbox. This is because the sub-surface is modified to represent the thermal characteristics of ice. No infiltration is allowed, and all melt is assumed to be surface runoff. The surface temperature is limited to the melting point with the residual energy balance term assumed to be melt. As such ice surfaces do not conserve water.

The roughness lengths for inland water, bare soil & ice were updated to their current values as part of GL4.0 (Walters, et al., 2014). The roughness length (*z0* – Table 1) for inland water was reduced to $1 \times 10^{-4}$ m as GL3.0 suffered from a slow bias





when compared to reanalyses in the near-surface wind speed around the Great Lakes. This reduced value is more consistent with the values predicted from wind-speed-dependent parametrisations over open water (Walters, et al., 2014). The roughness length for bare soil was increased to $1\times10^{-3}$ m, an intermediate value between those used in GL3.0 ($3\times10^{-4}$ m) and GL3.1 ($3.2\times10^{-3}$ m, used in operational global NWP forecasting). Observational estimates of the roughness length of bare

soil surfaces suggest large geographical variations covering this range. The ratios of the roughness lengths for heat to momentum *(z0hm* – Table 1) were also revised as part of GL4.0 in conjunction with the roughness length changes. From GL4.0 the urban surface has used the Best (2006) value of $1\times10^{-7}$ m; for inland water, the ratio has been set to 0.25 consistent with the parameterisation for open sea; bare soil was decreased to 0.02 to address a significant underestimate of the near-surface temperature gradient over arid regions; and ice was adjusted to 0.2 to be consistent with sea-ice. Prior to

GL4.0, all ratios had a fixed value of 0.1. Another new capability introduced with GL4.0 was an emissivity for each surface type (*emis* – Table 1) based on the data of (Snyder, Wan, Zhang, & Feng, 1998) and additionally for bare soil, satellite retrievals of land surface temperature from over the Sahara. Previously these values were fixed at 0.97 regardless of surface type. The significant reduction in the bare soil emissivity improved a cold bias over the Middle Eastern deserts that was prominent in GL3.0. The description of the non-vegetated surface types in JULES-GL7 remains largely unchanged since

GL4.0.Best et al., 2011)).

### 2.2 Radiation

Typically, standalone JULES is driven with downward short wave and long wave radiative fluxes. To obtain the net fluxes that enter the surface energy budget, the surface albedo and emissivity must be calculated. The albedo varies with wavelength, although, for many natural surfaces, it is adequate to distinguish between the visible and near infrared parts of the spectrum.

In reality, the albedo is also different for direct and diffuse radiation, but a distinction is not made for every surface in GL7.

For unvegetated surfaces single broadband albedos are used (*albsnf*, Table 1). The albedo of bare soil must be specified as ancillary data, but fixed values are used for the other three unvegetated tiles.

The albedo of plant canopies is calculated using the two-stream radiation scheme described by Sellers (1985). As inputs, this requires separate transmission (*omega, omnir*, Table 2) and reflection coefficients (*alpar, alnir,* Table 2) in the visible and near infrared regions respectively for individual leaves (or shoots in the case of needleleaf trees) and the leaf area index. It returns the visible and near infrared albedos for direct and diffuse radiation. However, the direct components are discarded and not used in GL7 for reasons of performance when coupled to the UM. When coupled to the UM, there is an option

*(l_albedo_obs)* to scale the leaf-level characteristics to match a specified climatology of the albedo, but this unavailable offline.

A new albedo scheme for snow-covered surfaces was introduced into GL7. This incorporates a two-stream algorithm for the snow pack. The surface is modelled as an underlying soil surface, above which there is a plant canopy that is gradually buried as snow accumulates. The canopy is therefore modelled as a lower snow layer and an upper layer of exposed vegetation that

will be absent if the snow is deep enough. The scheme makes explicit use of the canopy height. If canopy snow is allowed on the tile, there will also be a layer of snow on the canopy that is treated using the same two-stream scheme. Additional parameters (*can_clump, n_lai_exposed*) represent the vertical distribution of leaf-area density and the clumping of snow on the canopy. However, the values adopted in GL7 have been tuned to work with the existing ancillaries of canopy height which exhibit unrealistically limited spatial variability. Previously, a hard-wired lower limit of 0.5 had been imposed on the LAI in

the calculation of the albedo. At GL7 this has been removed and replaced with separate limits for snow and snow-free conditions. In snow-free conditions, the nominal lower limit has been set to 0.005, while in the presence of snow a limit of 1.0



is imposed for trees and a limit of 0.1 in the case of short vegetation (Table 2). Infrared emissivities are specified as single broadband values for each surface type (Tables 1 and 2).

### 2.2.1 Diffuse Radiation

GL7 assumes that photosynthetically active radiation (PAR) is half of total downwelling shortwave. The PAR as seen in the plant physiology is entirely direct, which results in a lower penetration of PAR into the canopy and reduced photosynthesis at the sub-canopy level. For GL7.2 we introduce a constant global mean diffuse fraction of 0.4, based on output from the SOCRATES radiative transfer scheme (Edwards & Slingo, 1996; Manners et al., 2015). This has the impact of increasing light penetration into the canopy and therefore increasing GPP. To further improve GPP, we updated the canopy radiation model

(*can_rad_mod*, changed from 4 to 6). Like *can_rad_mod*=5, 6 introduces sunfleck penetration through the canopy ($fsun = \exp(-\frac{k_b}{\cos z}LAI)$; $k_b$ *constant value of* 0.5) which increases the light within the canopy particularly for high solar zenith angles. Furthermore, *can_rad_mod=6* introduces a new nitrogen profile through the canopy following $\exp(-k_{nl}LAI)$, where $k_{nl}$ is a PFT constant of 0.2 (Table 2). This has the effect of increasing potential GPP in canopies with low LAI (< 5) and decreasing at high LAI (>5). GL7.2 is consequently a physically more realistic configuration of JULES. The changes to canopy

radiation do not affect the simulated albedo as in the current setup the albedo is only calculated for direct radiation. GL7.0 and 7.2 will therefore have the same albedo, but the way light interacts with the canopy differs and therefore affects the exchange of moisture and carbon.

### 2.3 Surface Exchange

The representation of the surface energy budget in JULES is described by Best et al. (2011). The scheme includes a surface heat capacity. Atmospheric resistances are calculated using standard Monin-Obukhov surface layer similarity theory, using the stability functions of Beljaars and Holtslag (1991). Evaporation from bare soil and water on the canopy and transpiration through the plant canopy contribute to the latent heat fluxes. In the case of needle-leaved trees, snow on and beneath the canopy is treated separately (*can_mod=4*).

### 2.4 Soil Hydrology and Thermodynamics

Soil processes are represented using a 4-layer scheme for the heat and water fluxes with hydraulic relationships taken from van Genuchten (1980). The four layers (0.1, 0.25, 0.65 and 2m) are chosen to capture diurnal, seasonal and multiannual variability in soil moisture and heat fluxes. The parameter values used in the scheme are described in Dharssi et al., (2009) and are read from an ancillary. There is an additional deep layer with impeded drainage to represent shallow groundwater thus

enabling a saturated zone and water table to form. The sub-grid scale soil moisture heterogeneity model is driven by the statistical distribution of topography within the grid box, and is based on a TOPMODEL-type approach (Gedney et al., 2003). The baseflow out of the model is dependent on the predicted grid box mean water table while surface saturation and wetland fractions are dependent on the distribution of water table depth within the grid box. The scheme uses the Marthews et al., (2015) topographic index dataset at 15 arc-sec resolution, which in turn is derived from HydroSHEDS (Lehner et al., 2006).

The soil and hydrological ancillaries required are listed in Table 3.



## 2.5 Snow

A major difference between GL7 and earlier GL configurations is the activation of the multilayer snow scheme in JULES that is described by Best et al. (2011). This replaces the previous so-called zero-layer scheme in which a single thermal store was used for snow and the first soil level, and an insulating factor was applied to represent the lower thermal conductivity of snow.

The zero-layer scheme included no representation of the evolution of the snow pack. Compared to the version described in Best et al. (2011), a number of enhancements have been introduced into the multilayer scheme in order to better to represent the thermal state of the snow surface and atmospheric boundary layer when coupled to the UM. The changes are noted in the following description and parameter values in Table 2.

In the multilayer scheme, the snow pack is divided into a number of layers, that are added or removed as the snowpack grows or shrinks. A maximum of 3 layers is imposed in GL7. In a deep snow pack, the top layer will be 0.04 m thick, the second 0.12 m thick, while the lowest layer will contain the remainder of the snow pack. Very thin layers of snow (less than 0.04 m deep) are still represented using the zero-layer scheme for reasons of numerical stability. In the original version of the scheme, the top two layers were set to default values of 0.1 and 0.2 m thick. The thickness, frozen and liquid water contents, temperature

and grain size of each layer are prognostics of the scheme. New snow is added to the top of the snow pack and compaction by the overburden is included. Following these operations, the snow pack is relayered to the specified thickness.

The density of fresh snow has been set to 109 kg m$^{-2}$, following the scheme adopted in the CROCUS model (Vionnet et al., 2012), but omitting the wind-speed and temperature-dependent factors. The conductivity of snow was originally calculated

using the parameterization of Yen (1981), but this has been replaced with the scheme proposed by Calonne et al. (2011). This gives higher conductivities in snow of low density, thereby strengthening the coupling between the snow pack and the boundary layer.

Again, with a view to improving the coupling between the atmosphere and the snow pack, the parameterization of

equitemperature metamorphism described by Dutra et al. (2010) has been introduced. This accelerates the rate of densification of fresh snow and is important in reducing cold biases that would otherwise result.

In the original scheme, when the canopy snow model was selected, unloading of snow from the canopy occurred only when it was melting. In GL7, unloading (*unload_rate_u*, Table 2) is also permitted at colder temperatures and the timescale is set to

1/(2.31e-6*U10), which is tuned to give an unloading timescale of 2 days in the Canadian boreal forest in winter (MacKay and Bartlett, 2006) for the average 10 m wind speed predicted in the MetUM. Note that a separate canopy is currently used only for the needleleaf tile.

Unlike the original scheme, where it simply bypassed the snow pack, rain water is now allowed to infiltrate. Below a canopy,

this infiltrating water includes melting from the canopy.

## 2.6 Coupled versus Uncoupled Differences

JULES has been developed in more than one modelling environment i.e. standalone and coupled with the UM, and consequently some science options are not available under all environments. This could be because certain science options

only make sense in a coupled environment or the converse may be true. This is not true for all options and in some cases the options have only been implemented in one environment and require additional coding to make it available to others. Other



differences arise out of the method of coupling the available driving data. When coupled, the surface meteorological state is solved interactively whereas offline this is provided either from observation or reanalysis products. One important difference concerning the treatment of radiation (*jules_radiation*) is that when coupled separate radiative fluxes of NIR and PAR are available from the radiation scheme however, offline typically only broadband shortwave is available and it is assumed this

can be split 50:50 between NIR and PAR. Furthermore, when coupled, snow-free albedos on each surface type are nudged towards an observed climatological mean from an ancillary (*l_albedo_obs = .true.*). This approach maintains sensible differences between surface types and allows spatial differences in albedo properties to be captured, while agreeing well with observations. However, in turn this has some limitations and as such it is not suitable for climate change experiments that include a change in land cover. It is therefore not compatible with the dynamic vegetation and land-use models as used in the

interactive carbon cycle option and thus is not implemented in the offline JULES-GL7 configuration. Another subtle difference concerning the treatment of radiation is the calculation of the solar zenith angle. When coupled, the SOCRATES radiative transfer scheme calculates this, whereas in standalone JULES the solar zenith angle calculation needs to be explicitly turned on using *l_cosz = .true.* to be equivalent. When using JULES-GL7 therefore with site data, care should be taken to ensure that the model and forcing data is in Coordinated Universal Time (UTC), as time is used in the calculation of solar zenith angle.

There are several differences in the treatment of the JULES surface exchange (*jules_surface*) as this is the interface between the surface and either the driving model or the driving data. Orographic form drag (*formdrag = 0 standalone, 1 coupled*) for example cannot be used in the standalone configuration as the necessary ancillary data is not available to standalone. In any case, it may not make scientific sense to include as the orographic drag may be implicit in the driving data whether from

observations or from model generated driving data. The method of discretization in the surface layer is another difference between the two environments, which affects how the driving data is interpreted. The driving data, when standalone, is most likely to be at a specific level (*i_modiscopt = 0*) rather than a vertical average as it is when coupled (*i_modiscopt = 1*). Also in the coupled model, a parametrisation of transitional decoupling in very light winds is included in the calculation of the 1.5 m temperature (*iscrntdiag =2*), however in standalone the surface is driven by the temperature at 1.5 m and is therefore is not a

diagnostic. It is not recommended that the surface is driven with a decoupled variable as this scenario has not been properly tested and should instead be *iscrntdiag = 0*. And finally concerning the surface exchange, the coupled model includes the effects of both boundary layer and deep convective gustiness (*isrfexcnvgust = 1*), however this is not appropriate in standalone and therefore *isrfexcnvgust = 0*. When driving standalone JULES with observations at a high enough frequency the gusts would be implicit in the observational data; and in the case of driving JULES with a longer-term average, where there may be

a gust contribution, the relevant information is not accessible.

### 3 JULES-GL7 Experimental Setup and Suite Control

The science configuration consists of a defined set of parameters and switches that can be used in conjunction with an experimental setup. The experimental setup differs from the configuration as it describes the conditions under which the

configuration is applied. For example, in this case the setup is a global historical experiment, but it could also be a future climate scenario, driven by alternative historical forcing or at a multiple locations such as FluxNet sites, where more detailed evaluation data are available (e.g. Harper et al., 2016). The experiment in the suite provided is a global historical run from pre-industrial (1860) to present day (2014) including rising atmospheric $CO_2$ but fixed landcover. This is a standard historical experimental setup as used in the Global Carbon Project (Le Quéré et al., 2015). The climate data (CRU-NCEP v7) consists

of 6-hourly NCEP data corrected to CRU climatology and observations updated to 2014 (CRU TS3.23; Harris et al., 2015). The original data were provided on a 0.5° × 0.5° grid and subsequently regridded to a coarser resolution for consistency with





the standard resolution climate experiments for CMIP6 using HadGEM3-GC3.1 at 1.875° × 1.25°. The forcing data include both gridded observations of climate and global atmospheric $CO_2$, which change over time (Dlugokencky and Tans, 2015). However, the CRU-NCEP data only start in 1901. To begin the experiments in 1860, a time when atmospheric $CO_2$ was relatively stable, requires the years 1901-1920 to be replicated between 1860 and 1900 thus assuming no effect of climate

5 change between 1860 and 1901. CRU-NCEP uses a 365-day calendar so no leap years are included. Furthermore, CRU-NCEP is a land-only dataset including Greenland but excluding Antarctica. At the coarser resolution, a gridbox may only be partially land-covered. JULES works on the land only fraction of the gridbox. It is therefore important when making global means or averages that both the land fraction of a gridbox as well as the gridbox area is considered. An important provided ancillary is therefore the land fraction (Table 3).

The suite as provided, includes a standardised suite control approach to manage both the necessary stages of initialising and running an experiment, as well as scheduling resources and timeslots on the supercomputer. This is shown graphically in Figure 4. The suite is set up to run three separate instances of JULES. The first initialises and reconfigures an initial start condition. The second starts from the reconfigured start condition and spins up the states of snow, soil moisture and

15 temperatures by cycling over 1860-1879 climate using fixed pre-industrial $CO_2$. This is optional according to whether the initial state is already spun-up and is controlled by a switch (*l_spinup)*. Setting this switch to false bypasses spinup entirely. The number of cycles required and the period to loop over can also be set. Each new cycle of spinup is submitted as a new job taking the initial conditions from the end of the previous cycle. The final task is to perform the transient experiment taking either initial conditions from the reconfiguration step or the final spinup cycle. These settings are all available under *Runtime*

*Configuration* and *Runtime Configuration > Spinup Options.* The transient run makes uses of varying climate and atmospheric $CO_2$. As standard the transient experiment has a 10 year cycle interval to allow a complete cycle to complete within the time limits on the supercomputer. It is worth noting that JULES vn5.3 is unable to perform full bit comparable restarts. This means the model prognostics at the end of one submission differ slightly from those used at the start of the next. The exact state at the end of the transient run will therefore be dependent on the number of spinup and transient cycles used.

## 4 JULES-GL7 Evaluation

In this section we evaluate the CRU-NCEPv7 historical experimental setup of the JULES-GL7 model configuration. We follow the approach of the International Land Model Benchmarking (ILAMB) project tool (Collier et al, 2018) to compare model simulations against observational data. However, due to technical limitations we are unable to use the full benchmarking range

that ILAMB includes. Here, we assess model performance against three key metrics covering surface energy balance, hydrology and vegetation productivity. The metrics are annual mean albedo, evapotranspiration and gross primary productivity and they are benchmarked against observationally based datasets available in ILAMB. The aim here is not to perform a full analysis of model skill but to establish a few important benchmarks against which model developments can be compared and evaluated. In time, it is planned that the standardised JULES suite will be fully compatible with ILAMB allowing for a full

model evaluation and benchmarking to be completed in a straight forward and standardised way. Furthermore, caution should be taken in benchmarking a model using a single forcing data set. As part of the Land Surface, Snow and Soil Moisture Model Intercomparison Project (LS3MIP; van den Hurk et al., 2016) this configuration will be setup with GSWP3 forcing data, which in time will be made available to the community. A second dataset will allow sampling of model uncertainty arising from forcing data variation.





Surface albedo is simulated in the model as described in Section 2.2 Radiation. Globally the observed land surface albedo is generally higher in snow-covered regions and deserts as shown in the MODIS satellite data (Figure 5). As noted in Section 2.2.1 Diffuse Radiation, the simulated albedo in JULES-GL7.0 and 7.2 are exactly the same, despite having different canopy radiation options, as the differences only affect light availability for photosynthesis. Overall, we find the model is too bright with a globally positive bias (Table 4). However, Figure 6 shows that the bias is spatially variable with the largest biases (both positive and negative) found in the high latitudes and other snow-covered regions. In general, in this experimental setup we find the surface is too bright in regions of boreal forests and too dark across the far north in the tundra regions.

Evapotranspiration is benchmarked against two observational products: GLEAM (Miralles et al., 2011) and MODIS (Mu et al., 2013). There is uncertainty in the two datasets with large differences in the magnitude of evapotranspiration particularly over the tropical regions (Figure 7). Both GL7 and 7.2 have large positive biases over much of the world, and these are strongest over the tropics (up to 2mm per day; Figure 8). However, the exact location of the largest biases differs between MODIS and GLEAM. GLEAM suggests a dipole pattern over central Africa while MODIS has a centralised positive bias implying there is a degree of observational uncertainty that needs to be accounted for. Overall, the biases are slightly reduced in GL7.2 (Table 4).

Although GL7 is mainly intended for studying the exchange of momentum, heat and water, the configuration also underpins the carbon-cycle configuration, and photosynthesis is strongly linked to evapotranspiration through the stomatal conductance model. It is therefore worth benchmarking the model's ability to simulate GPP. Here we compare simulated GPP against the Fluxnet-MTE product (Figure 9; Jung et al., 2010). Figure 10 shows that GL7.0 and 7.2 correctly predict that GPP is highest in tropical forests and low in arid areas, but there is a substantial negative bias in most biomes with the exception of tropical forests. GL7.2 is an improvement over GL7.0 with a global total GPP of 95.4 GtC compared with 91.1 GtC in GL7.0. However, this is substantially lower than the 119 GtC in the reference dataset.

## 5 Running the JULES-GL7 Setup

### 5.1 Compute Platform Setup

The JULES-GL7.0 and JULES-GL7.2 configurations are available as rose suites at https://code.metoffice.gov.uk/trac/roses-u/browser/b/b/3/1/6/trunk and https://code.metoffice.gov.uk/trac/roses-u/browser/b/b/5/4/3/trunk respectively. Note, access will be required to the Met Office Science Repository Service (https://code.metoffice.gov.uk/trac/home) and is available to those who have signed the JULES user agreement. JULES is freely available for non-commercial research use as set out in the JULES User Terms and Conditions (http://jules-lsm.github.io/access_req/JULES_Licence.pdf). The easiest way to access the repository is by completing the online form here (http://jules-lsm.github.io/access_req/JULES_access.html).

The suite is configured to run on both the Met Office CRAY XC40 or the JASMIN (http://www.jasmin.ac.uk/) platform provided by the Science and Technology Facilities Council UK. For non-Met Office collaborators JASMIN is the most suitable platform for running JULES simulations. JASMIN access is available for all UK based researchers who consider themselves part of the NERC (https://nerc.ukri.org/) community. JASMIN is also available for non-UK based researchers who are interested in JULES. Once you have access to JASMIN you will need to request access to the JULES group workspace (/group_workspaces/jasmin2/jules), which can be requested here


https://accounts.jasmin.ac.uk/services/group_workspaces/jules/. Met Office CRAY XC40 users will need access to the xcel00 and/or xcef00 machines.

Installing the suite requires access to the Met Office suite and code management tools available on both JASMIN and the Met Office Linux estate. To access the tools please follow the guidelines in Appendix A. Once you have access to the necessary compute platforms, repository and tools you are ready to start your run.

The suite is designed for ease of use, to enable the maximum number of users to access it. The suite is configured to extract the code from the repository, build on the appropriate platform sourcing appropriate libraries and then run using the appropriate

forcing and ancillaries. Most users should be able to set a standard run going in just a few steps.

**5.2 Setting up the model configuration**

The standard JULES-GL7.0 (*JULES-GL7.2*) suite, u-bb316, (*u-bb543*) has been configured to minimise the steps necessary to be able to run the standard configuration, however a few important steps and checks remain. It is assumed that a JASMIN user has logged into the jasmin-cylc node and a Met Office user is accessing the CRAY via a Linux desktop.

1.  Create a new suite:

    *rosie copy u-bb316*

    This will create a new suite of your own in which changes can be made and tracked using the Met Office Science Repository Service. Remember to commit any changes back to the repository with '*fcm commit.*' *Rosie copy u-bb316* results in a new suite with a similar id in alphanumeric order e.g. *u-ab123*. You should replace '*u-ab123*'

with your suite id in the following commands.

2.  The *rosie copy* command will create a local copy of the new suite in the *~/roses* directory. You can change directory to this suite.

3.  Once the suite is installed you can use the Rose GUI editor to check the suite setup. There are a number of platform specific aspects to be checked. To open the GUI:

    *rose edit -C ~/roses/u-ab123/*

    a.  Platform specific > Build and run mode – This radar button is used to setup the platform specific build and installation. This should be '*Met Office-cray-xc40*' and '*Jasmin-Lotus*' on the CRAY and JASMIN platforms

respectively.

    b.  Build options > JULES_FCM – This variable points to the location of the code to be compiled. In the standard case this should point to the trunk, however this could equally point to a branch to test a new development. An important point to note is that the CRAY uses an internal 'mirror' copy of the repository held in the cloud. This avoids downtime when the repository is unavailable. This is indicated by an '*m*' in

the repository shortcuts. This should be *fcm:jules.xm* and *fcm:jules.x* on the CRAY and JASMIN respectively. Failure to set this correctly will result in a build failure.

    c.  Runtime Configuration > MPI_NUM_TASKS – Up to 16 MPI tasks are available on JASMIN. More are available on the CRAY for faster run times. 16 MPI tasks is a recommended setup. However, 18 MPI (with 2 OMP) makes a fuller user of a single broadwell node on the CRAY.

40    d.  Runtime Configuration > OMP_NUM_TASKS – More recently releases of JULES support more OpenMP threads. A suitable number of tasks is 2.



The suite is now installed and ready to run. On the CRAY the submission can be made from the local machine. On JASMIN use the cylc workflow machine *jasmin-cylc*. The suite can be submitted to the scheduler.

*rose suite-run -C ~/roses/u-ab123/*

4.   Assuming the suite submits correctly, the next step is to monitor progress. Met Office and JASMIN users will automatically see the *suite control GUI*. However, the suite can be monitored by one of the two following options:

| | |
|---|---|
| *cylc scan –c* | Will show the state of running suites |
| *tail -f ~/cylc-run/u-ab123/log/suite/log* | Will print to screen the current status of u-ab123 |

5.   The output from the suite is automatically written to a directory:
a.   *$DATADIR/jules_output/u-ab123* on the CRAY
b.   */work/scratch/$USER/u-ab123* on JASMIN.
Note the scratch workspace on JASMIN is not for permanent storage of model output.

**5.3 Making changes to model configuration**

The purpose of making a standard science configuration and experimental setup available is not so users can reproduce the same results, but to encourage further development and testing, whether that involves new and novel diagnostics and evaluation or new processes and ancillary information. This should be done relative to the 'benchmark' standard configuration and experimental setup. To modify the configurations users should copy the standard suite as above and switch the code base to
point to the user's branch and revision number. Any new parameters and switches can then be added to the app configuration file – this can be done through the GUI or by editing the configuration file directly (*~/roses/u-ab123/app/rose-suite.conf)*. Note the model code needs to be consistent with the setup in the app. Any modifications to the suite should be committed and documented on a JULES ticket similar to the one documenting the JULES-GL7 release (https://code.metoffice.gov.uk/trac/jules/ticket/837).

**5.4 Inter-version compatibility**

The JULES-GL7.x model configurations are independent of the code release as it is a requirement of any modification to the JULES code base that the major configurations are scientifically reproducible between code versions. This is not exactly the same as reproducible to the bit level as some changes are permitted, for instance changing the order of a do loop can have benefits for runtime, but lead to changes are the bit level. From a user perspective the differences between model releases
should be pragmatically indistinguishably. It is intended that the JULES-GL7.x configurations will be made available at each model release and the latest release is preferable if undertaking configuration development. Users of the configuration may find benefits in the latest version through technical improvements to suite control tools including user interfaces and code optimisation reducing run time. It is therefore preferable to use the latest available configuration. At some point when a configuration is deemed superseded the guarantee of backwards compatibility will be dropped and code modules may be
removed from the code base and no longer supported.

**6 Summary**

JULES-GL7.0 is the standalone version of the land surface configuration underpinning the HadGEM3-GC3.1 climate model that is being run as part of the CMIP6 round of global climate modelling experiments. It is a comprehensive model simulating





the exchange of heat, water and momentum developed as part of the coupled climate model and extracted here for use by the community.

It has been shown that both JULES-GL7.0 and JULES-GL7.2 can capture the large-scale features of surface albedo,
5   evapotranspiration and GPP, however there are substantial biases that future updates to the configuration should attempt to reduce. There is also substantial uncertainty in observational evaluation datasets and the forcing for driving the model (Collier et al. 2018), which remains to be accounted for. Caution therefore needs to be taken to avoid overfitting the model to just a few datasets without a full appreciation of the uncertainties involved. In time, we plan to add additional forcing datasets to the standard configuration and the ability to benchmark against the full capability available in ILAMB.

This configuration and the ability to run the model is provided to the land surface modelling community to promote community engagement in the advancement of land surface science whether through application in their individual study, for use in model intercomparison studies such as LS3MIP (van den Hurk et al., 2016) or to promote community science developments progressing onto the main JULES trunk and into the major science configurations that underpin weather and climate
forecasting in the UK.

**Author Contributions**

AW coordinated the preparation of the JULES version of the coupled GL7 suite and manuscript. JE led the initial development and testing of the GL7 configuration. AW, CDR, and KSD undertook the technical development to make the configuration
available via JASMIN and the standardised suite control. AW, CDR, JE, NG, ABH, AH, MH and ER all prepared sections of the manuscript. All authors contributed to the preparation of the manuscript.

**Acknowledgements**

We thank Nicolas Viovy for providing the forcing data. This work was funded under the BEIS and DEFRA Met Office Hadley
Centre Climate Programme (MOHCCP) for 2018 – 2021. AW and CDR acknowledge the support of the EU Horizon 2020 CRECENDO project (grant agreement No 641816).

**Data availability**

The model configuration and associated forcing data are available via the indicated methods in the manuscript. JULES and associated configurations are freely available for non-commercial research use as set out in the JULES User Terms and
Conditions (http://jules-lsm.github.io/access_req/JULES_Licence.pdf).

**Code availability**

This work is based on JULES version 5.3 with specific configurations included in the form of suites. For full information regarding accessing the code and configurations, please refer to section 5.1



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





**Tables**

Table 1: Parameters in JULES-GL7 that vary with non-vegetated surface types (note these can be found in nvegparm).

|  | Urban | Lake | Bare Soil | Ice |
|---|---|---|---|---|
| **albsnf**<br>*Snow-free albedo* | 0.18 | 0.12 | -1 (indicates read *from ancillary*) | 0.75 |
| **catch**<br>*Water capacity (kg m⁻²)i* | 0.5 | 0 | 0 | 0 |
| **ch**<br>*Heat capacity of this surface type (J K⁻¹m⁻²).* | 280000 | 21100000 | 0 | 0 |
| **emis**<br>*Surface emissivity* | 0.97 | 0.985 | 0.9 | 0.99 |
| **gs**<br>*Surface conductance (m s⁻¹).* | 0 | 0 | 0.01 | 1000000 |
| **infil**<br>*Infiltration enhancement factor* | 0.1 | 0 | 0.5 | 0 |
| **vf**<br>*Switch indictaing whether the canopy is conductivity coupled (0) to the sub-surface or radiatively (1).* | 1 | 1 | 0 | 0 |
| **z0**<br>*Roughness length for momentum (m)* | 1 | 0.0001 | 0.001 | 0.0005 |
| **z0hm**<br>*Ratio of the roughness length for heat to the roughness length for momentum.* | 1E-07 | 0.25 | 0.02 | 0.2 |

5    Table 2: Parameters in JULES-GL7 that vary by PFT (note these can be found in pftparm and snow namelists).

|  | Broadleaf Tree | Needleleaf Tree | C3 Grass | C4 Grass | Shrub |
|---|---|---|---|---|---|
| **a_wl**<br>*Allometric coefficient relating the target woody biomass to the leaf area index* | 0.65 | 0.65 | 0.005 | 0.005 | 0.1 |
| **a_ws**<br>*Woody biomass as a multiple of live stem biomass.* | 10 | 10 | 1 | 1 | 10 |
| **albsnc_max**<br>*Snow-covered albedo for large leaf area index.* | 0.25 | 0.25 | 0.6 | 0.6 | 0.4 |





| | | | | | |
|---|---|---|---|---|---|
| **albsnc_min**<br>*Snow-covered albedo for zero leaf area index.* | 0.3 | 0.3 | 0.8 | 0.8 | 0.8 |
| **alnir**<br>*Leaf reflection coefficient for NIR* | 0.45 | 0.35 | 0.58 | 0.58 | 0.58 |
| **alpar**<br>*Leaf reflection coefficient for PAR (photosynthetically active radiation)* | 0.1 | 0.07 | 0.1 | 0.1 | 0.1 |
| **alpha**<br>*Quantum efficiency (mol $CO_2$ per mol PAR photons).* | 0.08 | 0.08 | 0.08 | 0.04 | 0.08 |
| **b_wl**<br>*Allometric exponent relating the target woody biomass to the leaf area index* | 1.667 | 1.667 | 1.667 | 1.667 | 1.667 |
| **c3**<br>*c3/c4 photosynthetic pathway switch* | 1 | 1 | 1 | 0 | 1 |
| **can_struct_a**<br>*Canopy Structure factor* | 1 | 1 | 1 | 1 | 1 |
| **catch0**<br>*This is the minimum amount of water that can be held on the canopy (kg $m^{-2}$)* | 0.5 | 0.5 | 0.5 | 0.5 | 0.5 |
| **dcatch_dlai**<br>*Rate of change of canopy capacity with LAI (kg $m^{-2}$).* | 0.05 | 0.05 | 0.05 | 0.05 | 0.05 |
| **dqcrit**<br>*Critical humidity deficit (kg $H_2O$ per kg air).* | 0.09 | 0.06 | 0.1 | 0.075 | 0.1 |
| **dz0v_dh**<br>*Rate of change of vegetation roughness length for momentum with height.* | 0.05 | 0.05 | 0.1 | 0.1 | 0.1 |
| **emis_pft**<br>*Surface emissivity* | 0.98 | 0.99 | 0.98 | 0.98 | 0.98 |
| **eta_sl**<br>*Live stemwood coefficient (kg $Cm^{-1}$/(m2 leaf))* | 0.01 | 0.01 | 0.01 | 0.01 | 0.01 |
| **f0**<br>*Ratio of internal to atmospheric CO2 concentration at 0. Humidity deficit (CI / CA for DQ = 0)* | 0.875 | 0.875 | 0.9 | 0.8 | 0.9 |
| **fd**<br>*Scale factor for dark respiration.* | 0.015 | 0.015 | 0.015 | 0.025 | 0.015 |
| **fsmc_mod** | 0 | 0 | 0 | 0 | 0 |



| | | | | | |
|---|---|---|---|---|---|
| Switch for method of weighting the contribution that different soil layers make to the soil moisture availability factor fsmc. | | | | | |
| **glmin**<br>Minimum leaf conductance for $H_2O$ (m $s^{-1}$). | 0.000001 | 0.000001 | 0.000001 | 0.000001 | 0.000001 |
| **infil_f**<br>Infiltration enhancement factor | 4 | 4 | 2 | 2 | 2 |
| **kext**<br>Light extinction coefficient | 0.5 | 0.5 | 0.5 | 0.5 | 0.5 |
| **knl**<br>Parameter for decay of nitrogen through the canopy, as a function of LAI | 0.2 | 0.2 | 0.2 | 0.2 | 0.2 |
| **kpar**<br>PAR extinction coefficient | 0.5 | 0.5 | 0.5 | 0.5 | 0.5 |
| **lai_alb_lim**<br>Minimum LAI permitted in calculation of the albedo in snow-free conditions. | 0.005 | 0.005 | 0.005 | 0.005 | 0.005 |
| **nl0**<br>Top leaf nitrogen concentration (kg N/kg C). | 0.04 | 0.03 | 0.06 | 0.03 | 0.03 |
| **neff**<br>Scale factor relating Vcmax with leaf nitrogen concentration | 0.0008 | 0.0008 | 0.0008 | 0.0004 | 0.0008 |
| **nr_nl**<br>Ratio of root nitrogen concentration to leaf nitrogen concentration. | 1 | 1 | 1 | 1 | 1 |
| **ns_nl**<br>Ratio of stem nitrogen concentration to leaf nitrogen concentration. | 0.1 | 0.1 | 1 | 1 | 0.1 |
| **omega**<br>Leaf scattering coefficient for PAR. | 0.15 | 0.15 | 0.15 | 0.17 | 0.15 |
| **omnir**<br>Leaf scattering coefficient for NIR. | 0.7 | 0.45 | 0.83 | 0.83 | 0.83 |
| **orient** | 0 | 0 | 0 | 0 | 0 |
| **q10_leaf**<br>Q10 factor for plant respiration. | 2 | 2 | 2 | 2 | 2 |
| **r_grow**<br>Growth respiration fraction. | 0.25 | 0.25 | 0.25 | 0.25 | 0.25 |
| **rootd_ft** | 3 | 1 | 0.5 | 0.5 | 0.5 |



| | | | | | |
|---|---|---|---|---|---|
| *Parameter determining the root depth (m).* | | | | | |
| **sigl** <br> *Specific density of leaf carbon (kg C m⁻²leaf).* | 0.0375 | 0.1 | 0.025 | 0.05 | 0.05 |
| **Tlow** <br> *Lower temperature for photosynthesis (deg C).* | 0 | -5 | 0 | 13 | 0 |
| **Tupp** <br> *Upper temperature for photosynthesis (deg C).* | 36 | 31 | 36 | 45 | 36 |
| **z0hm_pft** <br> *Ratio of the roughness length for heat to the roughness length for momentum.* | 1.65 | 1.65 | 0.1 | 0.1 | 0.1 |
| **Snow Parameters** | | | | | |
| **can_clump** <br> *Clumping factor for snow in the canopy* | 1 | 4 | 1 | 1 | 1 |
| **cansnowpft** <br> *Canopy snow model switch* | .false. | .true. | .false. | .false. | .false |
| **lai_alb_lim_sn** <br> *Lower limit on permitted LAI in albedo with snow* | 1 | 1 | 0.1 | 0.1 | 0.1 |
| **n_lai_exposed** <br> *Shape parameter for exposed canopy with embedded snow* | 1 | 1 | 3 | 3 | 2 |
| **unload_rate_cnst** <br> *Constant canopy snow unloading rate (kg m⁻² s⁻¹)* | 0 | 0 | 0 | 0 | 0 |
| **unload_rate_u** <br> *Wind dependent canopy snow unloading rate (kg m⁻²s⁻¹snow per ms⁻¹ wind)* | 0 | 2.31E-06 | 0 | 0 | 0 |





**Table 3: Ancillary information as required in the JULES-GL7/7.2 Configurations. Required ancillary files cover parameter values that are either spatially or temporarily explicit necessary to define the science configuration. Additional ancillaries covering grid setup and forcing are used in the experimental setup.**

| File | Fields and Description |
|---|---|
| **Science Configuration** | |
| Landcover Fractions | frac: Spatial fractional cover of each landcover tile |
| Vegetation Function | canht: Canopy height for vegetation tiles |
| | lai: Monthly Leaf Area Index Climatology for vegetation tiles |
| Soil properties | albsoil. Average waveband spatial field |
| | b: exponent in soil hydraulic characteristics |
| | hcap: Dry heat capacity |
| | satcon: Saturated hydraulic conductivity |
| | sathh: van Genuchten soil hydraulic 1/alpha parameter |
| | smcrit: volumetric soil moisture critical point |
| | smsat:saturated volumentric soil mositire |
| | smwilt: volumetric soil moisture wilting point |
| Hydrology | timean: spatial mean in topographic index |
| | tisig: spatial standard deviation in topographic index |
| **Experimental Setup** | |
| Land Fraction | Land_frac: fraction of a gridbox that is land |





**Table 4: Tabulated measures of model performance against benchmarks. Global Means and Totals are calculated on the native grid of the observational and model grids accounting for fractional land coverage in the totals and weighting for irregular grid box sizes. Biases and RMSEs are calculated by regridding the observational data to the coarser model grid and calculating metrics where the observational and model data intersect.**

| | Global Means/Totals | Bias | RMSE |
|---|---|---|---|
| **MODIS Albedo (Dimensionless)** | | | |
| Benchmark | 0.20 | | |
| GL7.0 | 0.25 | 0.039 | 0.074 |
| **GLEAM Evapotranspiration (mm day$^{-1}$)** | | | |
| Benchmark | 1.29 | | |
| GL7.0 | 1.72 | 0.35 | 0.65 |
| GL7.2 | 1.70 | 0.33 | 0.62 |
| **MODIS Evapotranspiration (mm day$^{-1}$)** | | | |
| Benchmark | 1.57 | | |
| GL7.0 | 1.73 | 0.38 | 0.63 |
| GL7.2 | 1.71 | 0.36 | 0.62 |
| **Fluxnet-MTE Gross primary Productivity (gC m$^{-2}$ day$^{-1}$)** | | | |
| Benchmark | 119 GtC | | |
| GL7.0 | 91.1 GtC | -0.6 | 1.06 |
| GL7.2 | 95.4 GtC | -0.5 | 0.99 |





Figures

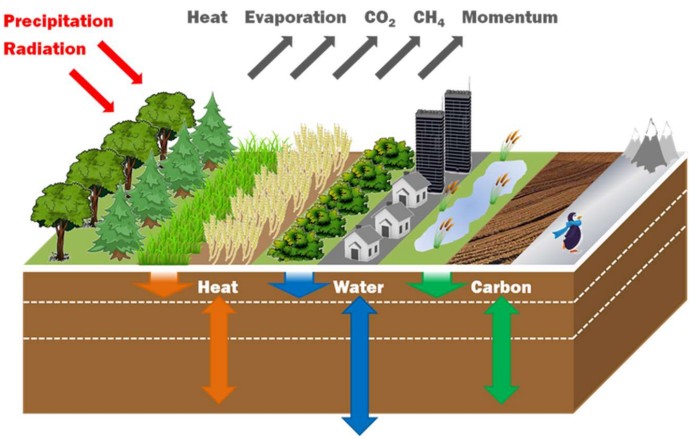

5    **Figure 1. JULES schematic of the fluxes of stores of heat, water, carbon and momentum and the surface tiling representation of subgrid heterogeneity.**

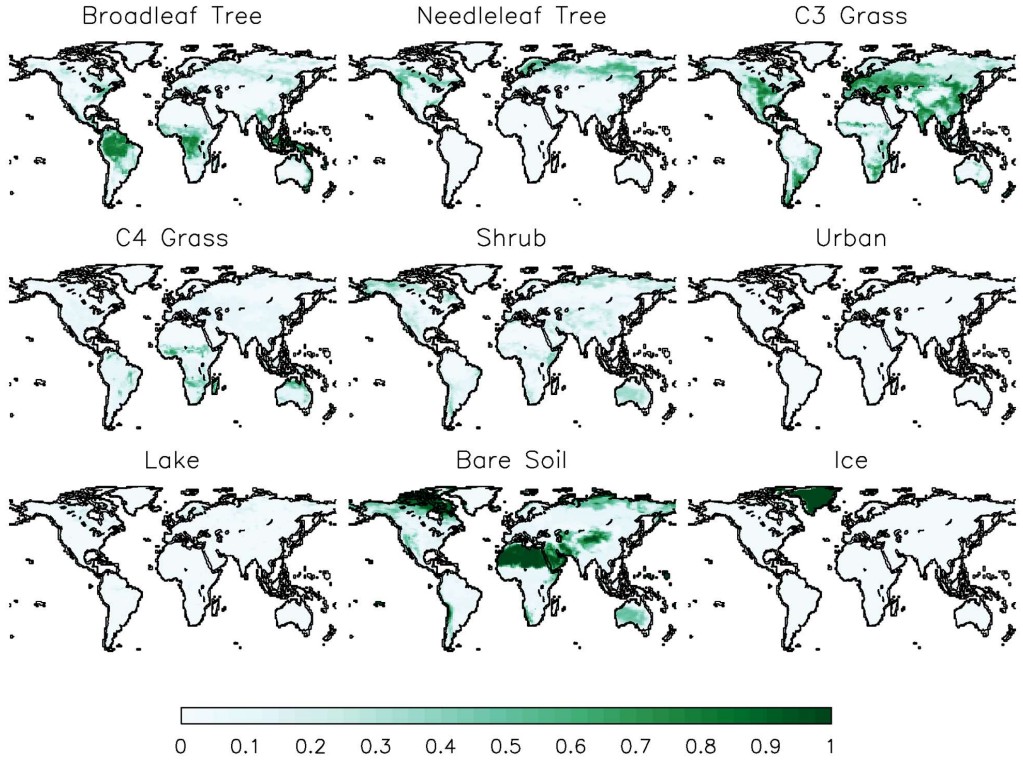

**Figure 2. Surface Tile fractions as used in JULES-GL7 derived from the IGBP landcover dataset (IGBP: Global Soil Data Task,**
10    **2000)**



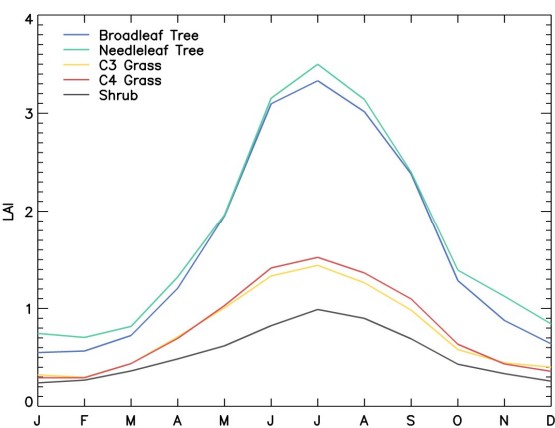

**Figure 3. Seasonal Leaf Area Index (LAI) for the 5 vegetation surface types area-averaged over 30-60°N.**

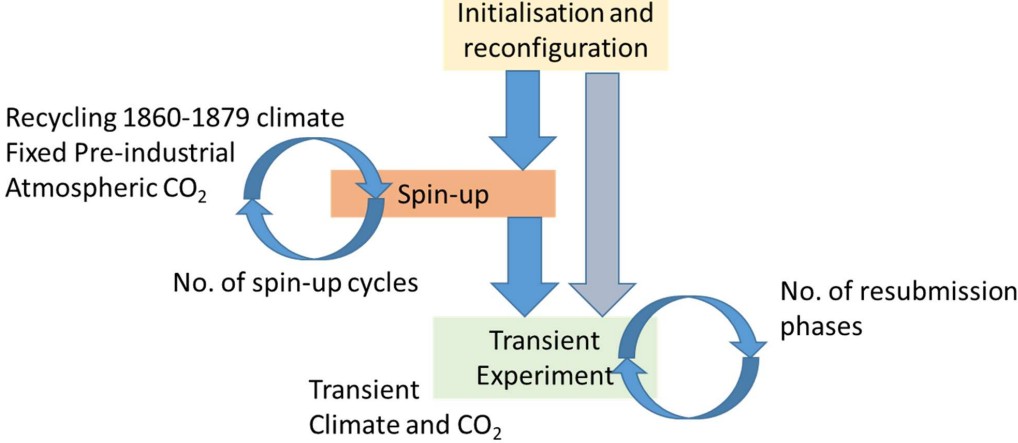

5    **Figure 4: Suite control used to initialise, spinup and perform a full transient experiment with JULES-GL7.**

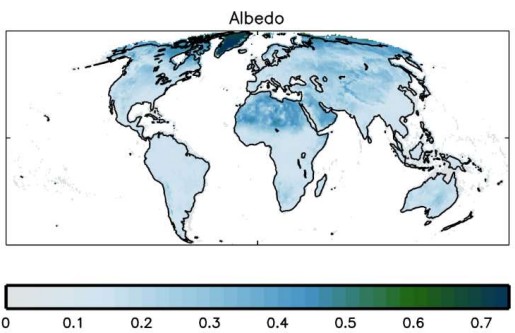

**Figure 5: Surface albedo 2000-2005 benchmark derived from MODIS (De Kauwe et al., 2011) as generated by ILAMB (Collier et al., 2018).**

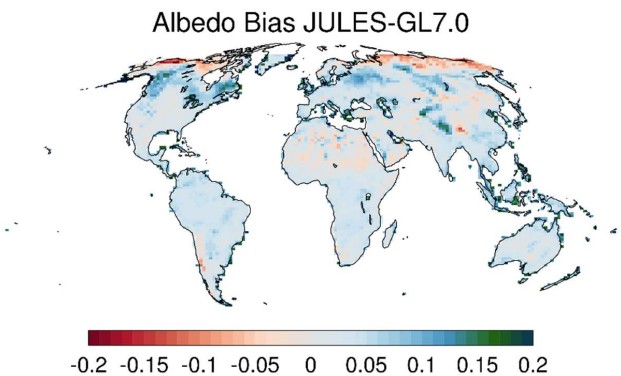

**Figure 6: Albedo bias simulated by GL7.0 relative to the MODIS benchmark. Means over 2000-2005 are shown. Biases are calculated as the difference between the model and observations.**

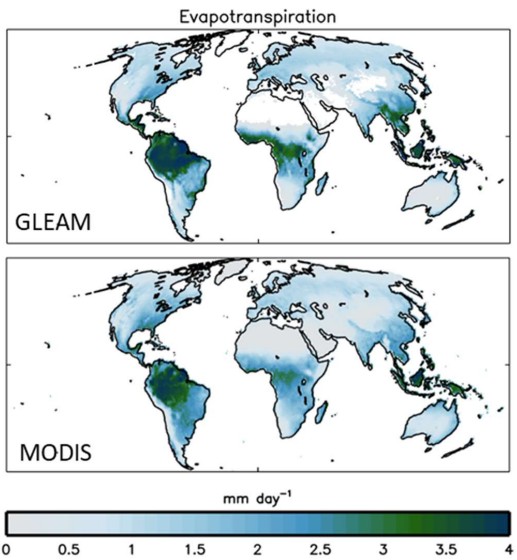

**Figure 7: Surface evapotranspiration benchmarks derived from GLEAM (Miralles et al., 2011) and MODIS (Mu et al., 2013) as generated by ILAMB (Collier et al., 2018) covering 1980-2011 and 2000-2013 respectively.**

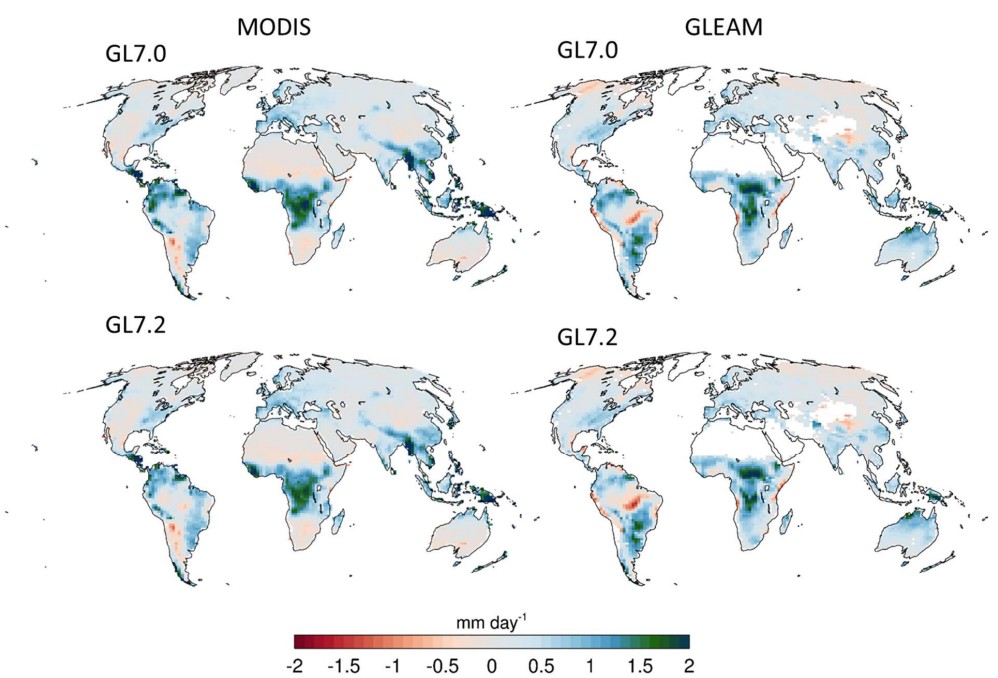

**Figure 8: Evapotranspiration biases simulated by GL7.0 (top row) and GL7.2 (bottom row) for MODIS (left column) and GLEAM**
5 **(right column) benchmarks. MODIS means are 2000-2013 and GLEAM 1980-2011. Biases are calculated as the difference between**
**the model and observations.**

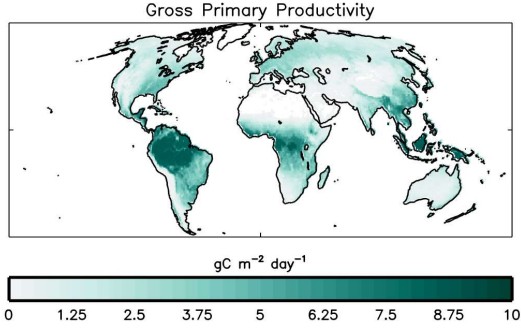

**Figure 9: Global Primary Productivity (GPP) (1982-2008) benchmark derived from Fluxnet- MTE (Jung et al., 2010) as generated**
10 **by ILAMB (Collier et al., 2018).**





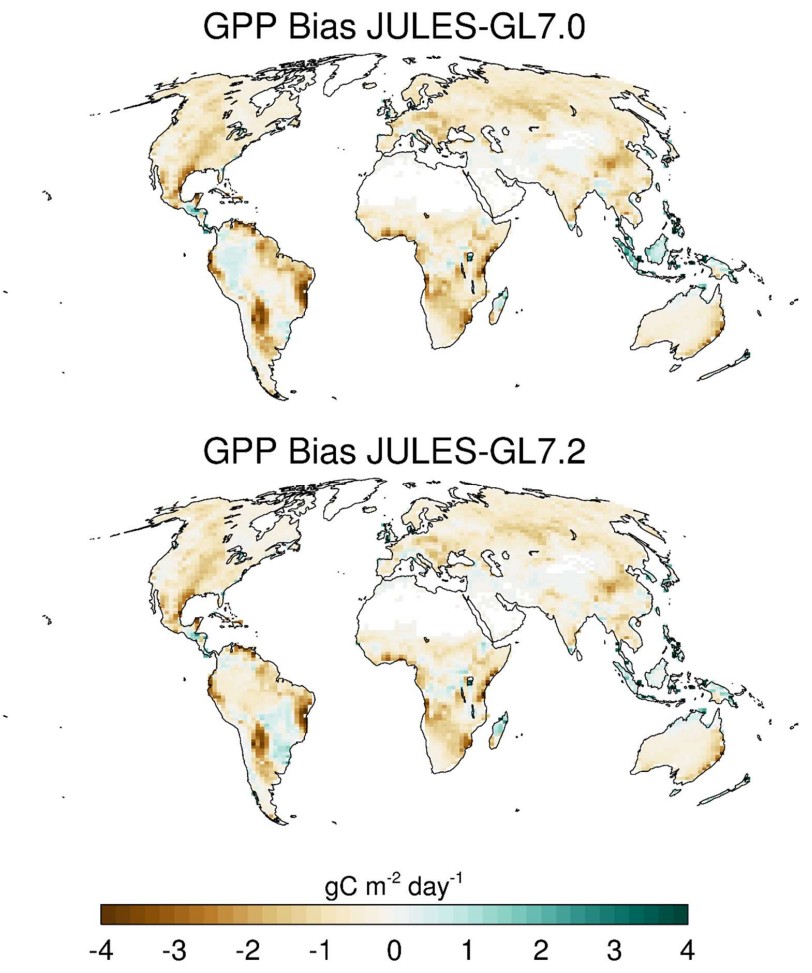

**Figure 10: Global Primary Productivity (GPP) biases simulated by GL7.0 (top row) and GL7.2 (bottom row) against the Fluxnet-MTE dataset. Means are 1982-2008. Biases are calculated as the difference between the model and observations.**





**Appendix A: Setting up the JASMIN work environment.**

The following assumes you have access to JASMIN as outlined in section 5. This section outlines the necessary steps to setup the necessary work environment.

5    On `jasmin-cylc`, edit your `~/.bash_profile` file:

```
# Get the aliases and functions

if [ -f ~/.bashrc ]; then
        . ~/.bashrc
fi

# User specific environment and startup programs

export PATH=$PATH:$HOME/bin
HOST=$(hostname)

if [[ $HOST = "jasmin-sci2.ceda.ac.uk" || $HOST = "jasmin-cylc.ceda.ac.uk" || $HOST = "jasm
in-sci1.ceda.ac.uk" ]]; then
  # Rose/cylc on jasmin-sci & Lotus nodes
  export PATH=/apps/contrib/metomi/bin:$PATH
fi
```

On `jasmin-cylc` edit your `~/.bashrc` file at the top:

```
# Provide access to FCM, Rose and Cylc
PATH=$PATH:/apps/contrib/metomi/bin

# Ensure .bashrc is sourced in login shells
# (only add this if it is not already done in your .bash_profile)
[[ -f ~/.bashrc ]] && . ~/.bashrc
```

At the bottom:

```
[[ $- != *i* ]] && return # Stop here if not running interactively
[[ $(hostname) = "jasmin-cylc.ceda.ac.uk" ]] && . mosrs-setup-gpg-agent
# Enable bash completion for Rose commands
[[ -f /apps/contrib/metomi/rose/etc/rose-bash-completion ]] && . /apps/contrib/metomi/rose/
etc/rose-bash-completion
```

Now, whenever login to `jasmin-cylc` you should be prompted for your Met Office Science Repository Service password.

40    A further setup for JASMIN and MOSRS requires an update to your `~/.subversion/servers` file. Please add the following and do not forget to give the corresponding username (change **myusername** word for your **MOSRS-username**).

```
[groups]
metofficesharedrepos = code*.metoffice.gov.uk

[metofficesharedrepos]
# Specify your Science Repository Service user name here
username = myusername
store-plaintext-passwords = no
```

50

In the `~/.subversion/config` file comment any lines starting with:





```
#password-stores =
```

Add the following lines on the `~/.metomi/rose.conf` file if missing (change **myusername** word for your **MOSRS-username**):

```
[rosie-id]
    prefix-default=u
    prefix-location.u=https://code.metoffice.gov.uk/svn/roses-u
    prefix-username.u=myusername
    #username is all in lower case
prefix-ws.u=https://code.metoffice.gov.uk/rosie/u

    [rose-stem]
    automatic-options=SITE=jasmin
```

This can be checked by running:

```
rose config
```





**Appendix B: Plant Functional Type, Leaf Area Index and Canopy Height Cross-Walking Tables**

**Table 5. PFT fraction lookup table for vegetated PFTs only. BLT = Broadleaf Tree; NLT = Needleleaf Tree. These lookup tables are used in conjunction with equations 1 and 2.**

|  | BLT | NLT | C3 Grass | C4 Grass | Shrub | Urban | Water | Bare Soil | Ice |
|---|---|---|---|---|---|---|---|---|---|
| Evergreen Needleleaf forest | 0 | 70 | 20 | 0 | 0 | 0 | 0 | 10 | 0 |
| Evergreen Broadleaf forest | 85 | 0 | 0 | 10 | 0 | 0 | 0 | 5 | 0 |
| Deciduous Needleleaf forest | 0 | 65 | 25 | 0 | 0 | 0 | 0 | 10 | 0 |
| Deciduous Broadleaf forest | 60 | 0 | 5 | 10 | 5 | 0 | 0 | 20 | 0 |
| Mixed forest | 35 | 35 | 20 | 0 | 0 | 0 | 0 | 10 | 0 |
| Closed shrub | 0 | 0 | 25 | 0 | 60 | 0 | 0 | 15 | 0 |
| Open shrub | 0 | 0 | 5 | 10 | 35 | 0 | 0 | 50 | 0 |
| Woody savannah | 50 | 0 | 15 | 0 | 25 | 0 | 0 | 10 | 0 |
| Savannah | 20 | 0 | 0 | 75 | 0 | 0 | 0 | 5 | 0 |
| Grassland | 0 | 0 | 70 | 15 | 5 | 0 | 0 | 10 | 0 |
| Permanent wetland | 0 | 0 | 80 | 0 | 0 | 0 | 20 | 0 | 0 |
| Cropland | 0 | 0 | 75 | 5 | 0 | 0 | 0 | 20 | 0 |
| Urban | 0 | 0 | 0 | 0 | 0 | 100 | 0 | 0 | 0 |
| Crop/natural mosaic | 5 | 5 | 55 | 15 | 10 | 0 | 0 | 10 | 0 |
| Snow and ice | 0 | 0 | 0 | 0 | 0 | 0 | 0 | 0 | 100 |
| Barren | 0 | 0 | 0 | 0 | 0 | 0 | 0 | 100 | 0 |
| Water bodies | 0 | 0 | 0 | 0 | 0 | 0 | 100 | 0 | 0 |

5   **Table 6. Leaf Area Index lookup table for combinations of IGBP land cover class and plant functional type. These lookup tables are used in conjunction with equations 1 and 2.**

|  | Broadleaf Tree | Needleleaf Tree | C3 Grass | C4 Grass | Shrub |
|---|---|---|---|---|---|
| Evergreen Needleleaf forest |  | 6 | 2 |  |  |
| Evergreen Broadleaf forest | 9 |  | 2 | 4 |  |
| Deciduous Needleleaf forest |  | 4 | 2 |  |  |
| Deciduous Broadleaf forest | 5 |  | 2 | 4 | 3 |
| Mixed forest | 5 | 6 | 2 |  |  |





| | | | 2 | | 3 |
|---|---|---|---|---|---|
| Closed shrub | | | 2 | | 3 |
| Open shrub | 5 | | 2 | 4 | 2 |
| Woody savannah | 9 | | 4 | | 2 |
| Savannah | 9 | | | 4 | |
| Grassland | | | 3 | 4 | 3 |
| Permanent wetland | 9 | | 3 | | 3 |
| Cropland | 5 | | 5 | 4 | 3 |
| Urban | | | | | |
| Crop/natural mosaic | 5 | 6 | 4 | 4 | 3 |
| Snow and ice | | | | | |
| Barren | | | | | |
| Water bodies | | | | | |

**Table 7. PFT-dependent canopy height scaling factor**

| | Broadleaf Tree | Needleleaf Tree | C3 Grass | C4 Grass | Shrub |
|---|---|---|---|---|---|
| Canopy height factor | 6.5 | 6.5 | 0.5 | 0.5 | 1.0 |

---
i