# Peer review of "JULES-GL7: The Global Land Configuration of the Joint UK Land Environment Simulation version 7.0 and 7.2"

_Geoscientific Model Development, 2019_

## Referee Comment (RC1) · Anonymous Referee #1 · 26 Jul 2019

The authors describe JULES-GL7, the latest land configuration for the JULES model. In particular JULES-GL7.0 and JULES-GL7.2 are the latest configurations for stan-dalone JULES (without an atmospheric model). The background to various configurations used by the UK Met Office is described, before the main part of the manuscript goes through each of the main areas of the model in turn, describing the approaches used. Later sections cover how to run the model and evaluation.

In general I think this is an important manuscript as it describes a key part of the modelling system, namely the configuration. The provision and description of configurations is an essential underpinning activity, on which an entire community of modellers

[Discussion paper]

[Figure]

can base their activities. Therefore I welcome this manuscript, although I have a few suggestions for relatively minor changes to the presentation. In particular I think that the main concepts of what a configuration is (and what it isn't) should be made clearer, and at any early point in the manuscript.

(Note that a large section of the manuscript is given over to describing how to run the model using standard suites on particular computers. I have not worked through the steps described and therefore cannot confirm their validity.)

Main comments

I would like to see a clearer explanation of exactly what defines a configuration, meaning what it covers and what it doesn't, how it differs from an "experimental setup" and the likes. There's a bit of this on P9 bottom paragraph, and possibly elsewhere, but it left me wanting to know more. Given that the paper is all about describing a configuration, it would be better to clarify the definition well before P9. Maybe at the start of section 2, when JULES configurations are first mentioned?

From what I understand, meteorological data are not part of the configuration, but I am less clear about some of the other inputs, e.g. soil and vegetation data. P7 L28 suggests some soil "parameter values...are described in...", and topographic index data are included; earlier we read about LAI derived from MODIS. Which of these are part of the configuration? Table 3 looks like the required inputs, but does not specify named files or sources for the information (e.g. MODIS processed according to a given recipe). It's all a bit confusing. If the datasets (or rather, input files derived from other datasets) are part of the configuration then (a) this should be clear, and (b) ideally we need to know more about the derivation of the files (thought that might be impractical to include). A diagram might be helpful here, to show what's in a configuration, and how it relates to other components of the system, e.g. the experimental setup, model suites, etc.. This might also be where the ideas of having standard model suites could also be explained.

[Figure]

P2 L17 and elsewhere: we are told that the paper covers GL7.0 and 7.2. In time we discover that these differ in terms of their treatment of radiation. It would be useful if the paper highlighted these differences more. For example, near P2 L17 briefly say that they differ slightly. And/or have a separate sub-section later that is just about GL7.2, so that the reader can easily navigate to find the answer as to how these configurations differ. And/or briefly note the differences at the top of Section 2. Also in abstract.

P2 L29 and elsewhere. I think the convention in use is that GL7 denotes a family of configurations, including GL7.0 and GL7.2. It would be good to have this clarified fromt he start, and to have the convention applied consistently - a.g. P3 L8 should be GL7.0? At present it is a bit confusing.

P3 L26: Here and elsewhere there are some terms that are probably more or less specific to JULES, e.g. ancillaries, rose suites, and that at any rate deserve more explanation for the broader readership. This is a wider point than just here - e.g. other comments about need to clarify what a "configuration" is. The ideas of suites etc. need to be properly introduced and woven into the manuscript at an appropriate place, and not assume too much background knowledge.

Section 4: There is a small amount of material related to evaluation of the configuration. The extent to which one paper can describe configurations and their evaluation is a tricky one, and it is important that the description of a configuration is not delayed substantially by the need to carry out a comprehensive evaluation. However I would suggest that any future updates on the JULES GL series should include a bit more on the evaluation, and/or signpost another set of papers that provides more in-depth evaluation.

Section 5: Most of this is very detailed and arguably is not required as part of the main manuscript (and for some people it will never be required). I suggest moving all or most of it to an Appendix. Only Sec5.4 "Inter-version compatibility" seems important enough for the main text, and I suggest that this should come much earlier as part of

the process of clarifying the terms and approaches used (the idea that the configuration is largely independent from the code version seems important to me, but at present the early discussion is possibly limited to a brief mention at P2 L36).

More minor comments and suggestions

Abstract: I would prefer to read more about the details of the GL7 configuration - at present the second half of the abstract is a rather rambling set of thoughts about the ideas behind the need for configurations, and similar. e.g. briefly note that GL7.0 and 7.2 are covered and how they differ?

Capitalise "coupled model intercomparison project"

Change "cluster accessible to all with links to JULES" to "cluster, accessible to all JULES users".

P2 L13: New paragraph at "JULES is the land component".

P2 L32 and nearby: Here I would just say that platforms and other tools are available, and give details later. Saying "Rose and Cylc" here doesn't add anything.

P2 L37 and others: Recommendations to use latest code version, temporary switches etc. - move these to later in the document. This level of detail is not useful this early, before we know much about the configurations themselves.

P2 L42 and others: There are many links that cannot be accessed without a valid login account. This should be indicated, e.g. with "login required". I suspect there might be a journal policy or guidance for this.

P4 L7: "Table B1" - inconsistent numbering. L12: add "Sections" before numbers.

P8 L13: "in the original version" - meaning what? An earlier configuration? An earlier iteration? I'm not sure we need to know this, and it shoudl certainly be made clearer.

Appendix A: JASMIN. This is very detailed information, and I worry that it might be the

kind of detail that tends to change relatively quickly as HPC platforms evolve. Could this detail be replaced by a reference to an online resource that is more likely to be kept up to date?

Appendix B: I'm pretty sure this is referenced before Appendix A - so change the order (B to become A).

Figures: In general I do not like colour schemes that use only 1 or 2 colours. They might look good but they tend to obscure information! e.g. Fig.2, Fig.5 (in particular!). However I realise these are very popular, so I will just note that they have major limitations!

---

## Short Comment (SC1) · 6 Aug 2019

Dear Professor Wiltshire

I would like to know why Jules users should use configuration GL7 rather than older configurations such as GL4. Please can you explain how GL7 compares to the other configurations that are described in the Jules documentation: http://jules-lsm.github.io/vn5.1/science-configurations.html

The Jules web-pages show some benchmarking for these older science configurations: https://jules.jchmr.org/sites/default/files/ILAMB%20Benchmark%20Results.png

[Figure]

How does GL7 skill compare? I couldn't find a similar table in the GL7 manuscript. Such a Table would help choose the best science configuration for my project.

The manuscript mentions Table B2, but it doesn't exist. Also I am confused whether the Jules canopy heights are remotely sensed or not. The Jules ancillaries I have looked at do not appear to use remotely sensed data. The numbers in Tables 6 and 7 look like "Guesstimates" rather than anything derived from observations or satellite measurements. Please can you provide plots of the Jules canopy heights.

---

## Short Comment (SC2) · 16 Sep 2019

**1   Soil Parameters**

P7L27: This sentence is not entirely accurate and better replaced by "The Jules-GL7 soil parameter values are based in part on soil parameter values developed for the MOSES model by Dharssi et al., (2009) and Cox et al., (1999)."

Dharssi et al (2009) is a Technical Report published by the UK Meteorological Office. The reference given in the discussion paper is unfortunately garbled.  The correct reference is:

[Figure]

Dharssi, I., Vidale, P. L., Verhoef, A., Macpherson, B., Jones, C. and Best, M.: New soil physical properties implemented in the Unified Model at PS18, Meteorology Research and Development Technical Report 528, Met. Office, Exeter, UK, [online] Available from http://research.metoffice.gov.uk/research/nwp/publications/papers/technical_reports/reports/528.pdf (Accessed 16 Sep 2019), 2009.

Cox, P. M., R. A. Betts, C. B. Bunton, R. L. H. Essery, P. R. Rowntree, and J. Smith. "The impact of new land surface physics on the GCM simulation of climate and climate sensitivity." Climate Dynamics 15, no. 3 (1999): 183-203.

**2 Section 2.1.1.1**

P4L33 is potentially misleading "The ancillaries are derived from satellite data processed ...". The Canopy heights are derived using parameters $h_i$ and $Lb_{i,j}$. Please can you clarify whether the PFT specific height scalar ($h_i$) and/or the balanced LAI ($Lb_{i,j}$) are derived from remote sensing data. The text references a non-existent Table B2, perhaps the correct reference is Table 6 or 7. Please can you clarify if Table 6 contains values for the balanced LAI and Table 7 contains values for the PFT specific height scalar. Spatial maps of Canopy height would allow the reader to more clearly judge the quality of the Jules Canopy heights and whether any remote sensing data has been used.

You might mention that GL9 uses remotely sensed tree heights and the work was in part influenced by
Dharssi, I., Steinle, P. and Fernon, J. 2015: Improved numerical weather predictions by using optimised urban model parameter values and satellite derived tree heights. 21st International Congress on Modelling and Simulation, Gold Coast, Australia. https://www.mssanz.org.au/modsim2015/M4/dharssi.pdf

**3   Table 3**

The entry "b: exponent in soil hydraulic characteristics" should be replaced by "b: van Genuchten soil hydraulic parameter 1/(n-1)".

---

## Referee Comment (RC2) · Anonymous Referee #2 · 2 Oct 2019

The paper submitted by Wiltshire et al. describe briefly the new JULES version from a scientific point of view but describes quite in details the configuration. I think this kind a paper is important to track the different version and help to increase transparency in Earth system modelling. The description of the simulations results are quite short but it is probably not so important for such paper. I suggest accepting the paper with some minor modifications: 1. The difference between GL7.0 and GL7.2 is not clearly explained and throughout the manuscript it was not so clear to me. Please explain briefly the differences in the introduction. 2. All the webpages started like https://code.metoffice.gov.uk need to a password and a login. Mentioning these webpages without providing access is not very useful. 3. There is a very limited description

of the C cycle so I guess you only represent GPP but what happens with the C fixed? This needs clarification 4. For eq. 2 where the allometric relationship comes from?

---

## Author Comment (AC1) · 8 Nov 2019

Thank you to the two reviewers and community members for considering our manuscript. Following recommendations we have restructured and clarified the role of configurations in JULES. We also restructure information regarding the access and use of the configuration to an Appendix. We include our responses inline below and new text in bold. We note the two reviewers considered only minor revisions were necessary. These have now been made and we hope you consider this ready for publication.

**Reviewer 1:**

*The authors describe JULES-GL7, the latest land configuration for the JULES model. In particular JULES-GL7.0 and JULES-GL7.2 are the latest configurations for standalone JULES (without an atmospheric model). The background to various configurations used by the UK Met Office is described, before the main part of the manuscript goes through each of the main areas of the model in turn, describing the approaches used. Later sections cover how to run the model and evaluation. In general I think this is an important manuscript as it describes a key part of the modelling system, namely the configuration. The provision and description of configurations is an essential underpinning activity, on which an entire community of modeller can base their activities. Therefore I welcome this manuscript, although I have a few suggestions for relatively minor changes to the presentation. In particular I think that the main concepts of what a configuration is (and what it isn't) should be made clearer, and at any early point in the manuscript. (Note that a large section of the manuscript is given over to describing how to run the model using standard suites on particular computers. I have not worked through the steps described and therefore cannot confirm their validity.)*

Thank you for the supporting comments and we agree this manuscript is a key aspect of the JULES numerical modelling ecosystem. Following your major comment on 'defining what a configuration is' we now include a new section in introduction and clarify throughout.

*Main comments I would like to see a clearer explanation of exactly what defines a configuration, meaning what it covers and what it doesn't, how it differs from an "experimental setup" and the likes. There's a bit of this on P9 bottom paragraph, and possibly elsewhere, but it left me wanting to know more. Given that the paper is all about describing a configuration, it would be better to clarify the definition well before P9. Maybe at the start of section 2, when JULES configurations are first mentioned? From what I understand, meteorological data are not part of the configuration, but I am less clear about some of the other inputs, e.g. soil and vegetation data. P7 L28 suggests some soil "parameter values...are described in...", and topographic index data are included; earlier we read about LAI derived from MODIS. Which of these are part of the configuration? Table 3 looks like the required inputs, but does not specify named files or sources for the information (e.g. MODIS processed according to a given recipe). It's all a bit confusing. If the datasets (or rather, input files derived from other datasets) are part of the configuration then (a) this should be clear, and (b) ideally we need to know more about the derivation of the files (thought that might be impractical to include). A diagram might be helpful here, to show what's in a configuration, and how it relates to other components of the system, e.g. the experimental setup, model suites, etc.. This might also be where the ideas of having standard model suites could also be explained.*

We take this comment on board and introduce a new section as part of the introduction '1.2 JULES Configurations.' As part of this we distinguish between science configuration, experimental setup and suite control and the broader aims and objectives of community model development.

Defining JULES configurations and how they should be used and developed is therefore an essential component of improving land surface modelling. At the core of an application is the science configuration, which is the collection of parameters, ancillaries and switches necessary to produce the same results for a given experimental setup. The experimental setup covers the necessary model forcing information to produce a simulation. For example, the setup provided here uses historical meteorological information to perform a simulation from the pre-industrial to present day at n96 (1.875º x1.25º) resolution.  Alternative experimental setups may be running future scenarios such as those included in CMIP6 (Eyring et al., 2016). The third component provided here is a standardised way by which the science and experimental configuration can be setup and run and is largely provided to support ease of access and use by a diverse range of users. This done by way of a suite compatible with the Rose/Cylc suite control system (https://metomi.github.io/rose/doc/html/index.html) available on JASMIN. The control system orchestrates the flow of inter-dependent tasks (workflow) from the initial extraction of the source code from repository, subsequent build and installation of the science and experimental setups and finally controls the simulation on the compute platform. The suite is the collection of all the information to make a simulation from start to finish in a format compatible with the workflow manager and a user-friendly graphical user interface.

JULES is a configurable model in which a named set of values control the operation of the model. JULES as a code base can support a number of these value sets that define different configurations. An important concept in the development of JULES-GL configurations is the independence of configuration from code release. JULES is managed to ensure that new developments in the code base produce scientifically comparable results. This is not exactly the same as reproducible to the bit level as some changes are permitted for example technical changes to the code base that result in explainable bit level changes. From a user perspective the differences between model releases should be pragmatically indistinguishable for a given configuration. The easiest way to ensure this, is for new developments to be put onto a switch. JULES-GL7 will be available at subsequent model versions and tested to ensure the setup produces scientifically comparable results between model code base versions until a date when JULES-GL7 is superseded and retired.  A second concept is that JULES as a code base can support multiple configurations dependent on the desired application. The two major configurations are Global Land and Earth System. The Earth System extends the Global Land to include biogeochemical processes important to understanding feedbacks in the climate system.

We consider the release of GL7 as the first step in a process to developing a comprehensive community based modelling approach. This will include comprehensive benchmarking and evaluation tools and a documented system for producing ancillary model information.  The aim is to further enhance community engagement in the development and improvement in the standard configurations that underpin UK national capability in weather, climate and hydrological modelling.

*P2 L17 and elsewhere: we are told that the paper covers GL7.0 and 7.2. In time we discover that these differ in terms of their treatment of radiation. It would be useful if the paper highlighted these differences more. For example, near P2 L17 briefly say that they differ slightly. And/or have a separate sub-section later that is just about GL7.2, so that the reader can easily navigate to find the answer as to how these configurations differ. And/or briefly note the differences at the top of Section 2. Also in abstract.*

Agreed. Now covered in introduction

*P2 L29 and elsewhere. I think the convention in use is that GL7 denotes a family of configurations, including GL7.0 and GL7.2. It would be good to have this clarified fromt he start, and to have the convention applied consistently - a.g. P3 L8 should be GL7.0? At present it is a bit confusing.*

Agreed. Now covered in introduction.

*P3 L26: Here and elsewhere there are some terms that are probably more or less specific to JULES, e.g. ancillaries, rose suites, and that at any rate deserve more explanation for the broader readership. This is a wider point than just here - e.g. other comments about need to clarify what a "configuration" is. The ideas of suites etc. need to be properly introduced and woven into the manuscript at an appropriate place, and not assume too much background knowledge.*

Agreed. This is now covered in the new section as part of the introduction.

*Section 4: There is a small amount of material related to evaluation of the configuration. The extent to which one paper can describe configurations and their evaluation is a tricky one, and it is important that the description of a configuration is not delayed substantially by the need to carry out a comprehensive evaluation. However I would suggest that any future updates on the JULES GL series should include a bit more on the evaluation, and/or signpost another set of papers that provides more in-depth evaluation.*

We appreciate the evaluation material could be further developed. Indeed as part of future updates we plan to develop a comprehensive benchmarking and evaluation system with GL7.0 as the initial benchmark. This would enable the impact of a change in configuration to be clearly documented in terms of its impact against the base setup. However, this is very much part of an ongoing activity.

*Section 5: Most of this is very detailed and arguably is not required as part of the main manuscript (and for some people it will never be required). I suggest moving all or most of it to an Appendix. Only Sec5.4 "Inter-version compatibility" seems important enough for the main text, and I suggest that this should come much earlier as part of the process of clarifying the terms and approaches used (the idea that the configuration is largely independent from the code version seems important to me, but at present the early discussion is possibly limited to a brief mention at P2 L36).*

Agreed. Most of section 5 is now in the appendix. As suggested we introduce the independence between configuration and code version in the introduction.

*More minor comments and suggestions*

*Abstract: I would prefer to read more about the details of the GL7 configuration - at present the second half of the abstract is a rather rambling set of thoughts about the ideas behind the need for configurations, and similar. e.g. briefly note that GL7.0 and 7.2 are covered and how they differ?*

*Capitalise "coupled model intercomparison project"*

Done

*Change "cluster accessible to all with links to JULES" to "cluster, accessible to all JULES users". P2 L13: New paragraph at "JULES is the land component".*

Done

*P2 L32 and nearby: Here I would just say that platforms and other tools are available, and give details later. Saying "Rose and Cylc" here doesn't add anything.*

Done

*P2 L37 and others: Recommendations to use latest code version, temporary switches etc. - move these to later in the document. This level of detail is not useful this early, before we know much about the configurations themselves.*

Agreed. Thinned section and moved some of the detail to appendix.

*P2 L42 and others: There are many links that cannot be accessed without a valid login account. This should be indicated, e.g. with "login required". I suspect there might be a journal policy or guidance for this.*

All links in the main documentation are now indicated where login is required. The number of links has also been reduced as part of the manuscript restructuring.

*P4 L7: "Table B1" - inconsistent numbering. L12: add "Sections" before numbers.*

Done

*P8 L13: "in the original version" - meaning what? An earlier configuration? An earlier iteration? I'm not sure we need to know this, and it shoudl certainly be made clearer.*

Superfluous and removed.

*Appendix A: JASMIN. This is very detailed information, and I worry that it might be the kind of detail that tends to change relatively quickly as HPC platforms evolve. Could this detail be replaced by a reference to an online resource that is more likely to be kept up to date? Appendix B: I'm pretty sure this is referenced before Appendix A - so change the order (B to become A).*

We have updated the appendix to include much of section 5 as suggested. We also note that significant effort has gone into the suite design and configuration of HPC platforms with the view of producing a long-term solution for community access to JULES. We feel this is an important part of the community offering. We include additional reference to where the living documentation can be found.

Appendix A was intended to be referenced in the introduction but was lost during editing. As part of the restructuring and clarification Appendix A is now referenced before B.

*Figures: In general I do not like colour schemes that use only 1 or 2 colours. They might look good but they tend to obscure information! e.g. Fig.2, Fig.5 (in particular!). However I realise these are very popular, so I will just note that they have major limitations!*

We note this for the future but keep the colours scheme in this manuscript.

**Reviewer 2:**

*The paper submitted by Wiltshire et al. describe briefly the new JULES version from a scientific point of view but describes quite in details the configuration. I think this kind a paper is important to track the different version and help to increase transparency in Earth system modelling. The description of*

*the simulations results are quite short but it is probably not so important for such paper. I suggest accepting the paper with some minor modifications:*

*1. The difference between GL7.0 and GL7.2 is not clearly explained and throughout the manuscript it was not so clear to me. Please explain briefly the differences in the introduction.*

Agreed. We now include this in the introduction and make clear on GL7 as a family of configurations as suggested by Reviewer 1.

*2. All the webpages started like https://code.metoffice.gov.uk need to a password and a login. Mentioning these webpages without providing access is not very useful.*

We appreciate the difficulties here but are limited in our options. We indicate a login is required now in the text.

*3. There is a very limited description of the C cycle so I guess you only represent GPP but what happens with the C fixed? This needs clarification*

We now make clear that GPP does not affect vegetation structure (LAI and canht) and is purely diagnostic in this setup. Furthermore, in the introduction we make it clear this is the non-biogeochemical configuration for physical climate simulation.

*4. For eq. 2 where the allometric relationship comes from?*

Jones, C.P.. Ancillary file generation for the UM. Unified Model Documentation Paper #73. Met Office Technical Documentation, 1998.

Now cited in the manuscript

**Interactive Comment 1:**

*I would like to know why Jules users should use configuration GL7 rather than older configurations such as GL4. Please can you explain how GL7 compares to the other configurations that are described in the Jules documentation: http://juleslsm.github.io/vn5.1/science-configurations.html The Jules web-pages show some benchmarking for these older science configurations: https://jules.jchmr.org/sites/default/files/ILAMB%20Benchmark%20Results.png*

Alternative examples of configurations have been previously been made available to the community as a stop-gap measure but the provenance of these is unknown and has resulted in a number of cases of poor performance. The release of JULES-GL7 is part of a national capability activity to ensure integrity of science results and enhance future development and capability of JULES land surface modelling. We do not recommend the use of these previous examples and work is underway to remove them from the standard examples and websites.

*How does GL7 skill compare? I couldn't find a similar table in the GL7 manuscript. Such a Table would help choose the best science configuration for my project.*

This would be very useful and the way we are moving. Unfortunately, I cannot answer that at the moment. However, GL4 is getting on for being 10 years old and uses code marked as deprecated.

*The manuscript mentions Table B2, but it doesn't exist. Also I am confused whether the Jules canopy heights are remotely sensed or not. The Jules ancillaries I have looked at do not appear to use remotely sensed data. The numbers in Tables 6 and 7 look like "Guesstimates" rather than anything*

*derived from observations or satellite measurements. Please can you provide plots of the Jules canopy heights.*

Tables 6 and 7 were mislabelled in the original manuscript. Canopy heights are a pragmatic combination of satellite information and scaling factors. Look out for a later iteration of GL in which canopy height will be derived from Lidar data. We now include a spatial plot of canopy heights in the manuscript.

**Interactive Comment 2:**

*P7L27: This sentence is not entirely accurate and better replaced by "The Jules-GL7 soil parameter values are based in part on soil parameter values developed for the MOSES model by Dharssi et al., (2009) and Cox et al., (1999)." Dharssi et al (2009) is a Technical Report published by the UK Meteorological Office. The reference given in the discussion paper is unfortunately garbled. The correct reference is:*

*Dharssi, I., Vidale, P. L., Verhoef, A., Macpherson, B., Jones, C. and Best, M.: New soil physical properties implemented in the Unified Model at PS18, Meteorology Research and Development Technical Report 528, Met. Office, Exeter, UK, [online] Available from http://research.metoffice.gov.uk/research/nwp/publications/papers/technical_reports/reports/528. pdf (Accessed 16 Sep 2019), 2009. Cox, P. M., R. A. Betts, C. B. Bunton, R. L. H. Essery, P. R. Rowntree, and J. Smith. "The impact of new land surface physics on the GCM simulation of climate and climate sensitivity." Climate Dynamics 15, no. 3 (1999): 183-203. 2*

Thankyou. Corrected in manuscript.  Note, we used a more up to date link for the Dharssi paper.

Section 2.1.1.1

*P4L33 is potentially misleading "The ancillaries are derived from satellite data processed ...". The Canopy heights are derived using parameters $h_i$ and $L_{b_{i,j}}$ . Please can you clarify whether the PFT specific height scalar ($h_i$) and/or the balanced LAI ($L_{b_{i,j}}$ ) are derived from remote sensing data. The text references a non-existent Table B2, perhaps the correct reference is Table 6 or 7. Please can you clarify if Table 6 contains values for the balanced LAI and Table 7 contains values for the PFT specific height scalar. Spatial maps of Canopy height would allow the reader to more clearly judge the quality of the Jules Canopy heights and whether any remote sensing data has been used.*

Fixed references to tables. Clarified in the text that canopy height is based on allometric scaling of landcover classes.

*You might mention that GL9 uses remotely sensed tree heights and the work was in part influenced by Dharssi, I., Steinle, P. and Fernon, J. 2015: Improved numerical weather predictions by using optimised urban model parameter values and satellite derived tree heights. 21st International Congress on Modelling and Simulation, Gold Coast, Australia.*
*https://www.mssanz.org.au/modsim2015/M4/dharssi.pdf*

We include mention that a priority for GL9 is to improve canopy height.

*Table 3 The entry "b: exponent in soil hydraulic characteristics" should be replaced by "b: van Genuchten soil hydraulic parameter 1/(n-1)".*

Thankyou. Clarified